# Direct Detection of Inhomogeneity in CVD-Grown 2D TMD Materials via K-Means Clustering Raman Analysis

**DOI:** 10.3390/nano12030414

**Published:** 2022-01-27

**Authors:** Hang Xin, Jingyun Zhang, Cuihong Yang, Yunyun Chen

**Affiliations:** 1School of Physics & Optoelectronic Engineering, Nanjing University of Information Science & Technology, Nanjing 210044, China; 20191217010@nuist.edu.cn (H.X.); yangcuihong1978@163.com (C.Y.); yunqq321@sina.cn (Y.C.); 2Jiangsu Key Laboratory for Optoelectronic Detection of Atmosphere and Ocean, Nanjing University of Information Science & Technology, Nanjing 210044, China; 3Jiangsu International Joint Laboratory on Meterological Photonics and Optoelectronic Detection, Nanjing University of Information Science & Technology, Nanjing 210044, China

**Keywords:** Raman, inhomogeneity, k-means cluster analysis, strain, doping

## Abstract

It is known that complex growth environments often induce inhomogeneity in two-dimensional (2D) materials and significantly restrict their applications. In this paper, we proposed an efficient method to analyze the inhomogeneity of 2D materials by combination of Raman spectroscopy and unsupervised k-means clustering analysis. Taking advantage of k-means analysis, it can provide not only the characteristic Raman spectrum for each cluster but also the cluster spatial maps. It has been demonstrated that inhomogeneities and their spatial distributions are simultaneously revealed in all CVD-grown MoS_2_, WS_2_ and WSe_2_ samples. Uniform p-type doping and varied tensile strain were found in polycrystalline monolayer MoS_2_ from the grain boundary and edges to the grain center (single crystal). The bilayer MoS_2_ with AA and AB stacking are shown to have relatively uniform p-doping but a gradual increase of compressive strain from center to the periphery. Irregular distribution of $2LA\left(M\right)/E_{2g}^{1}$ mode in WS_2_ and $E_{2g}^{1}$ mode in WSe_2_ is revealed due to defect and strain, respectively. All the inhomogeneity could be directly characterized in color-coded Raman imaging with correlated characteristic spectra. Moreover, the influence of strain and doping in the MoS_2_ can be well decoupled and be spatially verified by correlating with the clustered maps. Our k-means clustering Raman analysis can dramatically simplify the inhomogeneity analysis for large Raman data in 2D materials, paving the way towards direct evaluation for high quality 2D materials.

## 1. Introduction

Two-dimensional (2D) transition metal dichalcogenides (TMD), MX_2_ (M = Mo, W and X = S, Se, etc.), have attracted considerable attention due to their unique physical and chemical properties, exhibiting promising applications on optoelectronics [1,2,3], valley-electronics [4] and chemical sensors [5,6]. Both fundamental research and potential applications are highly dependent on the quality of TMD materials [7,8,9,10]. Conventionally, there are two kinds of methods to prepare 2D materials: top-down and bottom-up approaches. The typical top-down method, mechanical cleavage with scotch tape [7,8], could give high-quality mono- and few-layer TMD materials with limited size preventing their actual applications. On the other hands, bottom-up synthesis methods (such as intercalation assisted exfoliation [11], physical vapor deposition [12], hydrothermal synthesis [13], and chemical vapor deposition (CVD)) can offer a lateral size of TMD films up to hundreds of micrometers. In particular, CVD has been well-developed to produce large area crystals with controllable thickness and stacking sequences [9,10,14]. As a matter of fact, it is still a challenge to obtain highly uniform TMD materials with high performance on carrier mobility and conductivity with the CVD method. Due to the complicated growth processes in CVD and the interfacial interaction with substrate, mono- and few-layer TMD materials often host various inhomogeneities such as defects, strain and doping [15,16,17,18,19,20], which affects their mobility and electronic properties [21,22,23,24,25]. These common issues call for a direct characterization of local inhomogeneity in CVD-grown TMD materials.

Raman spectroscopy has been demonstrated as an effective and non-destructive method to characterize layered materials, such as graphene and TMD materials. The characteristic Raman bands of TMD materials represent the change of layer thickness [8,26], charge doping [27,28,29], defects [30,31], and strain [32,33,34]. Conley et al. observed the red-shifting of $E_{2g}^{1}$ mode from mono- to bi-layer MoS_2_ when applying uniaxial tensile strain [32]. Parkin et al. reported that the Raman characteristic mode shift can be used as a means to evaluate the sulfur vacancies in MoS_2_ [27]. Moreover, Chae W.H. and coworkers used the Raman spectral correlations to analyze the strain and doping in single-layer MoS_2_ grown by CVD method [35].

However, these studies were based on the Raman measurements at multiple points selected in a random way which is not applicable for samples with uniform optical contrast. The traditional Raman imaging is very useful for TMD material characterization [8,36]. However, it isn’t so efficient to do advanced spectral fitting, especially when dealing with thousands of Raman data of large size TMD. Another difficulty emerges when we try to decouple the strain and doping effects and further correlate them with sample areas. Therefore, it is essential to present a simple and efficient method to evaluate the material quality and probe the detailed inhomogeneity information.

As one of the most prevalent multivariate and unsupervised methods, k-means clustering is well-developed and has been applied to bio-Raman analysis [37,38]. The clustering is established from the spectral similarities reflecting all Raman spectral bands information, such as peak intensity, peak position, and peak width. In this work, we proposed an unsupervised k-means clustering Raman analysis [39] to directly identify the inhomogeneity in MoS_2_, WS_2_ and WSe_2_ samples grown by the CVD method. The whole spectral features of all Raman bands were selected for automatic clustering to keep the integrity. Based on the Raman spectral similarity, the clustered maps and several representative Raman spectra for each cluster could be obtained. The Raman spectral parameters in all clustered results are then correlated to evaluate the influences of strain and doping quantitatively in each clustered area. Varied strain-induced inhomogeneities and uniform p-type doping were observed in monolayer MoS_2_ with polycrystalline structure. With the help of k-means clustering, the bilayer MoS_2_ with AA and AB stacking were verified with four typical areas from center to periphery with a gradual increase of compressive strain and uniform p-doping. We also extended such analysis to probe the inhomogeneity in monolayer WS_2_ and WSe_2_. Non-uniform distribution of $2LA\left(M\right)/E_{2g}^{1}$ mode in WS_2_ and $E_{2g}^{1}$ mode in WSe_2_ is observed which is caused by strain. All the inhomogeneity mentioned above were directly visualized in color-coded Raman imaging with the correlated characteristic spectra. Combination of Raman spectroscopy with k-means clustering was proven to be efficient to distinguish the inhomogeneities distributions in the 2D materials caused by strain and doping. These results have demonstrated to be important for quality check in CVD-grown TMD and pave the way towards the high-performance TMD devices.

## 2. Experimental Section

MoS_2_, WS_2_ and WSe_2_ were grown by CVD on Si substrates with a 300 nm thick SiO_2_ [40]. Raman spectra and imaging were carried out using a confocal Raman microscope (WITec Alpha 300R, Ulm, Germany) equipped with the excitation source of 532 nm and a UHTS 300 spectrometer with 300 mm focal length. The laser beam was focused on the sample via 100 × 0.9 objective lens with the lateral spatial resolution better than 350 nm. The signal was collected through the same lens, dispersed with a 2400 grooves/mm grating for Raman spectra and was eventually detected by a FI-CCD camera with 1600 × 200 pixels (Andor) cooled to −60 °C. It could provide good spectral resolution 1.0 cm^−1^ and high repeatability better than 0.02 cm^−1^, which guarantees sufficient accuracy for the inhomogeneity analysis. The laser power was set to be 0.5 mW avoiding any damage on sample. All the Raman imaging data was classified by clustering and then statistically correlated via WITec Project Five software.

## 3. The K-Means Clustering and Inhomogeneity Visualization for Confocal Raman Analysis

The algorithm of k-means clustering aims at dividing the 2D, two-way data into the given *K* classes ($C_{1},\;C_{2},\;.\;.\;.\;,C_{k}$), where $C_{k}$ is the set of $n_{k}$ objects in cluster k. There are *N = I × J* total spectra and each one has a measurement of the *P* variable-spectral points in pixels in which *I* and *J* are the number of pixels or the number of the measurements along the *I* and *J*-coordinates. The centroid of the cluster $C_{k}$ is a point in the *P*-dimensional space, which is calculated by averaging the values of each variable on the objects in the cluster. For example, the centroid value of the jth variable in the cluster $C_{k}$ is
(1)$${\bar{x}}_{j}^{\left(k\right)}=\frac{1}{n_{k}}\underset{i\in C_{k}}{\sum }x_{ij},$$

And the complete centroid vector of cluster *C* is
(2)$${\bar{x}}^{\left(k\right)}={\left({\bar{x}}_{1}^{\left(k\right)},{\bar{x}}_{2}^{\left(k\right)},\ldots ,{\bar{x}}_{p}^{\left(k\right)}\right)}^{\prime }$$

The typical k-means clustering algorithm operates through the following iterative process:(1)The *P*-dimensional vector ($s_{1}^{\left(k\right)},\;\ldots \;,s_{p}^{\left(k\right)}$
) defines K initial seeds, for 1 < k < K, and the squared Euclidean distance, *d*^2^(i, k) (between the ith object and the kth seed vector) is obtained: $d^{2}\left(i,\;k\right)=\sum_{j=1}^{p}{\left(x_{ij}-s_{j}^{\left(k\right)}\right)}^{2}$.(2)Assign spectra to the smallest cluster. After initial Raman spectra allocation, the following operation is to obtain the cluster centroid for each cluster. All spectra are compared to each centroid (using *d*^2^(i, k)) and moved to the cluster whose centroid is the closest.(3)Use the updated cluster membership to calculate the new centroid (by calculating the centroids after all spectra have been assigned).(4)Steps 2 and 3 are repeated until each cluster no longer changes.(5)As illustrated in Figure 1a,b, 2D Raman spectra was collected via point scanning on TMD materials. In the 3D data set, there are *N = I × J* total spectra with *p =* 1600 spectral points in wavenumber unit. As shown in Figure 1c, in order to ensure the clustering accuracy, the Raman spectral range was particularly selected and the spectral points were reduced to $P^{\prime }$ (range of characteristic peaks of TMD materials). For example, all Raman spectra could be finally divided into three categories (cluster I, II, III) through k-means clustering as shown in Figure 1d. The cluster spatial maps and the correlated mean spectra are shown in Figure 1e. Such classification is based on the similarity of whole spectral features (peak intensity, peak position and width).

## 4. Results and Discussion

### 4.1. K-Means Raman Analysis of Monolayer MoS_2_

We probe inhomogeneity in monolayer CVD-grown MoS_2_ through the combination of Raman spectroscopy and k-means clustering, as shown in Figure 2. An optical image of the MoS_2_ sample is displayed in Figure 2a. Confocal Raman imaging was conducted on an 18 × 18 μm^2^ area with 200 nm step size and 8100 spectra were gained in total. It can be inferred that the overall frequency difference between $A_{1g}$(out-of-plane) and $E_{2g}^{1}$ (in-plane) modes is about 15~21 cm^−1^ as shown in Figure 2b which is consistent with the previous report on monolayer MoS_2_ [41]. On the other hand, we noticed that the $A_{1g}$ mode shows uniform distribution around 404 cm^−1^ only with slightly lower frequency along grain boundaries (GB) and sample edges as displayed in Appendix A. On the contrary, the $E_{2g}^{1}$ mode shows prominent red shift from GB and sample edge to grain center, as inferred from Appendix A, indicating irregular tensile strain distribution from GB to grain center. The $E_{2g}^{1}$ and $A_{1g}$ modes are often correlated with uniaxial strains [32,33] and charge doping [27,28], respectively. Hence, we can conclude that dominant tensile strain and slight doping condition coexist in this monolayer MoS_2_. The tensile strains could result from the different thermal expansion coefficients of the substrate during CVD growth [16]. However, the influence of strain and doping effects on peak intensities and peak width remains unknown. Moreover, such Raman measurements and analysis always require expertise and much effort to extract useful information, especially when dealing with tremendous Raman data in large size TMD. Here, we applied an efficient method to obtain a more comprehensive understanding of all these complicate images and check material quality directly.

We re-evaluated the Raman spectra using k-means clustering analysis, which could give a rapid classification between data points via squared Euclidean distance. The two characteristic Raman modes of MoS_2_,$\;A_{1g}$ and $E_{2g}^{1}$, were both selected and the Raman data were automatically analyzed. It turned out that the 8100 spectra were eventually classified into three clusters. The mean Raman spectra and the corresponding cluster spatial maps are shown in Figure 2d–f,h, respectively. The $E_{2g}^{1}$ modes of cluster 1–3 were located at 383.9, 384.6, and 385.3 cm^−1^, respectively, while the $A_{1g}$ mode was almost at 404 cm^−1^ through Lorentz function fitting, as shown in Appendix A. These cluster spectra could easily reveal the gradient trend of strain distributions in the whole sample, cluster 1 > cluster 2 > cluster 3, which is impossible to distinguish from E^1^_2g_ mode in the traditional Raman analysis in Appendix A. Obviously, the k-means clustering could classify Raman data and also dramatically reduce the spectral data analysis from thousands of spectra to a few ones. Meanwhile, by merging the cluster maps, the inhomogeneity could be easily recognized in a color-coded image, as shown in Figure 2g, providing direct visualization for the evaluation of sample quality.

The correlation analysis of Raman spectral features has been extensively utilized to decouple the strain and doping effect of graphene and TMD materials [42]. Michail et al. plotted a correlation graph between Pos($E_{2g}^{1}$) and Pos($A_{1g}$) to distinguish the strain and doping contributions in monolayer MoS_2_ prepared by mechanical exfoliation [43]. They manually allocated the sample area by optical contrast or conspicuous defect positions and then artificially subdivided the sample into several areas. Apparently, such manual selections are in-debate due to low precision. Moreover, it does not work for uniform optical images in which the sample inhomogeneity cannot be directly discerned, as shown in Figure 2a. Fortunately, the k-means method plays an important role in analyzing Raman spectral features of TMD materials with homogeneous optical contrast. The clustered correlation graphs were plotted as a function of peak position (Pos($E_{2g}^{1}$), Pos($A_{1g}$)) and peak width ($E_{2g}^{1}$) to further evaluate the strain and doping effects. For better comparison, the original and clustered Raman data were firstly displayed in 3D, as shown in Figure 3a,b, respectively. Obviously, the clustered distribution map between $E_{2g}^{1}$ and $A_{1g}$ peak positions clearly revealed more information about the inhomogeneity of MoS_2_ induced by strain and doping effects. By adopting the same method with Michail’s work, the distribution of strain and doping was further calculated by comparing their linear and independent influences on peak positions, as shown in Figure 3c. The ε ~ n (ε and n represent functions of strain and charge carrier concentration, respectively) relationships are calculated by choosing the Grüneisen parameters (0.86 for $E_{2g}^{1}$ and 0.15 for $A_{1g}$ modes) and the carrier concentration (−0.33 × 10^−13^ cm^−2^ for $E_{2g}^{1}$ and −2.22 × 10^−13^ cm^−2^ for $A_{1g}$ modes) [43,44,45]. The frequencies of $E_{2g}^{1}$ (384.7 cm^−1^) and $A_{1g}$ (402.7 cm^−1^) with 532 nm laser was selected as the unperturbed phonon frequencies in our calculations. The strain range in cluster 1–3 was 0.1~0.2%, 0~0.1% and 0.1~−0.1%, respectively. The overall p-doping concentration was about 0.5~1.0 × 10^−13^ cm^−2^, which may result from the absence of sulfur atoms and sulfur vacancies occupied by oxygen atoms during the CVD growth process [31].

### 4.2. K-Means Raman Analysis of Bilayer MoS_2_ with Different Stacking

It is well-known that the stacking order of TMD materials plays an important role in their physical and chemical properties [46,47,48]. Therefore, we further apply k-means cluster analysis to visualize the inhomogeneity of the CVD-grown bilayer MoS_2_ with different stacking. Confocal Raman imaging was conducted on a 12.5 × 12.5 μm^2^ area with 150 nm step size and 7225 spectra were gathered. Optical images of bilayer MoS_2_ with different stacking were displayed in Figure 4a,d, which were identified based on the relative intensity ratios of the breathing and shear modes in low wavenumber polarized Raman (as shown in Appendix A) [49]. The behavior of strain and doping inhomogeneity in the bilayer MoS_2_ with two different stacking are similar. Here, we only discussed the situation in triangular bilayers of MoS_2_ with AA stacking.

The k-means cluster analysis and corresponding Raman spectra are shown in Figure 4b,c,e,f. The samples were grouped into four regions (cluster 1, cluster 2, cluster 3, and cluster 4), as displayed in Figure 4b,e. Here we used the frequencies of $E_{2g}^{1}$ (382.5 cm^−1^) and $A_{1g}$ (404.9 cm^−1^) with 532 nm laser as the unperturbed phonon frequencies of bilayer MoS_2_. For AA stacking, the mean frequencies of the $E_{2g}^{1}$ mode of cluster 1–4 are 383.1, 383.6, 384.1 and 384.5 cm^−1^ as illustrated in Figure 4c.

We noted that the peak position of the $E_{2g}^{1}$ mode gradually shifted towards higher wavenumber from cluster 1 to cluster 4, indicating an increased compressive strain from center to periphery. The strain state of these two bilayer MoS_2_ samples is consistent with the results of Luo et al. [17]. This might be induced by different thermal expansion coefficients between the substrate and MoS_2_ during the CVD growth. There is no shift of $A_{1g}$ mode from cluster 1 to 4. Interestingly, we found the full width at half maximum (FWHM) of $A_{1g}$ mode becomes broader, indicating the increase of doping concentrations from cluster 1 to cluster 4. P-type doping MoS_2_ with AA stacking is evidenced by the overall blue shift of $A_{1g}$ mode compared with the unperturbed phonon frequencies in Appendix A. All in all, the clustered correlation analysis offers a solution to the cluster spatial map and further decouple and quantify the inhomogeneity of strain and doping effects.

### 4.3. K-Means Raman Analysis of Monolayer WS_2_ and WSe_2_

We further apply the k-means clustering Raman analysis to visualize the inhomogeneity in monolayer WS_2_ and WSe_2_. The optical images of triangular monolayer WS_2_ and WSe_2_ on Si/SiO_2_ substrate grown by the CVD method show a very homogeneous contrast, as illustrated in Figure 5a,d. Confocal Raman imaging of WS_2_ and WSe_2_ were conducted on a 20 × 20 μm^2^ area with 200 nm step size and 9775 and 9500 spectra were obtained, respectively. When excited with 532 nm laser, their resonance Raman spectra became much more complex, with several second-order Raman modes and a first-order prominent one as $2LA\left(M\right)/E_{2g}^{1}$ mode in WS_2_ and $E_{2g}^{1}$ in WSe_2_, as shown in Figure 5g,h. Their peak positions coincided with the previous Raman measurements, confirming their monolayer TMD structure [50,51]. The traditional Raman images of the prominent $2LA\left(M\right)/E_{2g}^{1}$ and $E_{2g}^{1}$ peak intensity for WS_2_ and WSe_2_ plotted in Figure 5b,e cannot manifest any inhomogeneities. With the help of k-means clustering, the local inhomogeneity can be clearly identified via the mean Raman spectra and then visualized through the color-coded Raman image. In the case of WS_2_, the triangle sample could be classified into two groups, cluster 1 and cluster 2, as displayed in Figure 5c. Comparing with cluster 1, it can be easily seen that the cluster 2 had higher intensity of the 2LA(M) to $E_{2g}^{1}$ as displayed in Figure 5g. It might have resulted from the local nanocrystallities formation with different sizes during CVD growth, as in the observation on single-layer WS_2_ treated by ion implantation by Tan, P.H. and his co-workers [52]. It is difficult to obtain more specific information of strain and doping, due to the complexity of the resonant excitation of WS_2_ under 532 nm. While in the monolayer WSe_2_ sample, the clustered results in Figure 5f,h exhibit three different areas with minor spectral shifting of $E_{2g}^{1}$ peak position, indicating the different level of strain during the CVD process (cluster 1 > cluster 2 > cluster 3). Another peak appeared on the shoulder of the $E_{2g}^{1}$ peak due to the effect of strain. The $E_{2g}^{1}$ peak split into two peaks, $E_{2g}^{+}$ and $E_{2g}^{-}$, representing the in-plane atomic vibrations in orthogonal and parallel ways, respectively. The spectral shifting of the $E_{2g}^{1}$ peak was similar to the result of Tang N. Y. et al. [53], which can be further confirmed as tensile strain. Remarkably, the k-means clustering Raman analysis could benefit for directly visualizing the existence of the inhomogeneity in TMD layered materials in a high accuracy and efficient way, providing important quality-control for TMD materials by the CVD method.

## 5. Conclusions

In this paper, we have successfully demonstrated that the combination of k-means clustering and Raman analysis was able to rapidly reveal the inhomogeneity distribution in the CVD-grown MoS_2_, WS_2_ and WSe_2_ layered materials. The k-means clustering method can provide the color-coded clustered Raman maps and corresponding mean Raman spectra. It could be beneficial to further identify the origin of various inhomogeneities. Introducing k-means analysis with correlations enables us to decouple and quantitatively evaluate the strain and doping contributions quantitatively in clusters. For the polycrystalline monolayer MoS_2_, there were obvious tensile strain and uniform p-type doping from the GB and edges to each grain center (single crystal). The bilayer MoS_2_ with AA and AB stacking was found with relatively uniform p-type doping and a gradual increase in compressive strain from center to periphery. In the case of monolayer WS_2_ and WSe_2_, we observed a non-uniform distribution of inhomogeneity of the $2LA\left(M\right)/E_{2g}^{1}$ mode and the $E_{2g}^{1}$ mode due to defect and strain, respectively. Our Raman analysis together with k-means clustering suggest that it is possible to directly detect the inhomogeneities and quantitatively evaluate the contributions of strain and doping in TMD materials. We expect that such automatic and unsupervised Raman analysis could be utilized to characterize the inhomogeneity and optimize TMD materials during the CVD process in order to obtain high performance devices.

## Figures and Tables

**Figure 1 nanomaterials-12-00414-f001:**
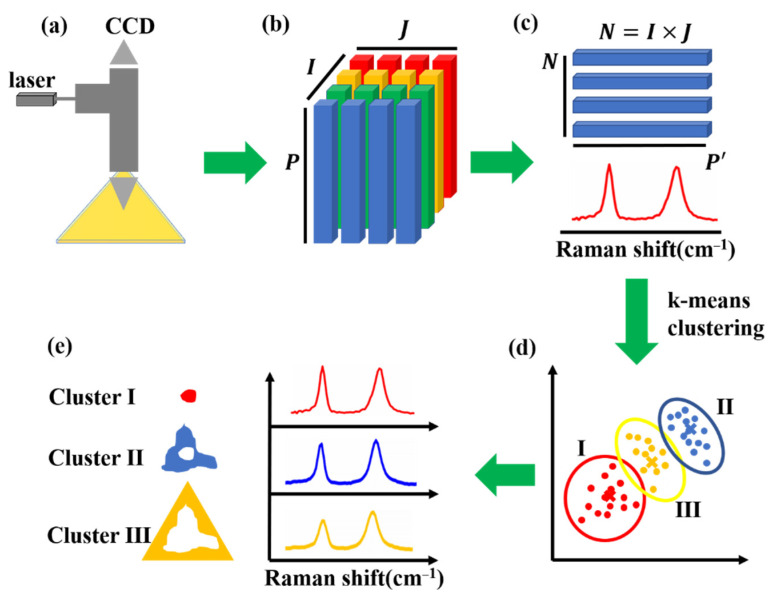
Schematic illustration of the workflow of inhomogeneity visualization in 2D TMD materials, from Raman imaging data acquisition to k-means clustering analysis. (**a**) Experimental setup for collecting Raman data of TMD materials. (**b**) 2D Raman imaging of TMD materials were collected *N = I × J* total spectra with *P =* 1600 spectral points in wavenumber unit shown in 3D way. (**c**) Define Raman spectral range with $P^{\prime }$ spectral points for k-means clustering analysis. (**d**) After running k-means clustering, the Raman data was automatically classified into cluster I, cluster II, and cluster III. (**e**) Color-coded cluster spatial maps and the correlated mean spectra of three clusters with identical color.

**Figure 2 nanomaterials-12-00414-f002:**
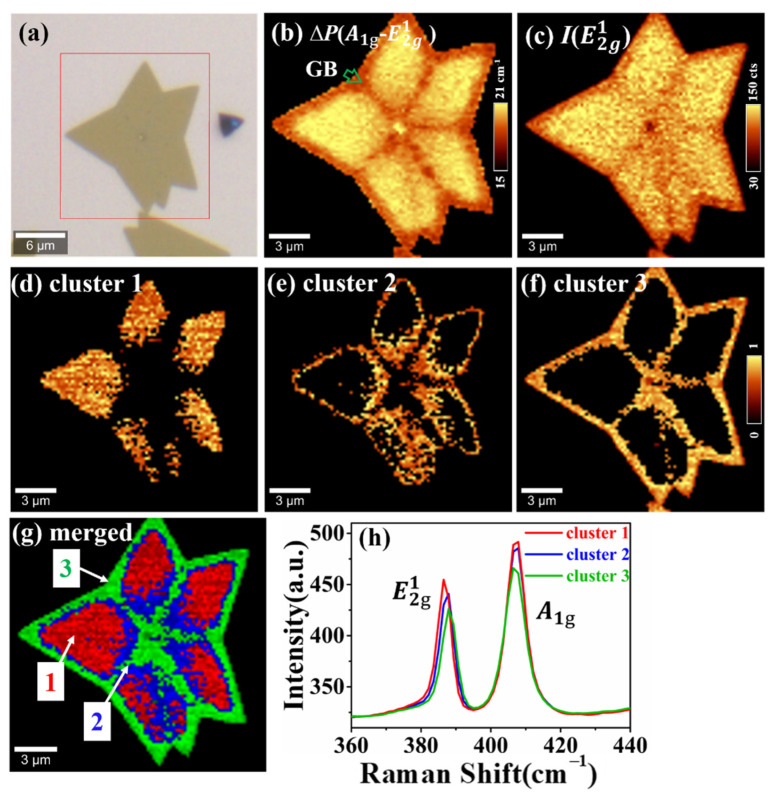
K-means clustering Raman analysis of inhomogeneity in monolayer polycrystalline CVD-growth MoS_2_ sample. (**a**) Optical image of monolayer polycrystalline MoS_2_ on a Si/SiO_2_ wafer. (**b**) Raman images of peak position differences between $A_{1g}$ and $E_{2g}^{1}$ modes. (**c**) Raman image of the absolute intensity of $E_{2g}^{1}$ mode. (**d**–**f**) Three clustered maps obtained by k-means clustering method with the correlated characteristic spectra. (**g**) The merged image of all clusters: Cluster 1 (red), cluster 2 (blue), and cluster 3 (green). (**h**) Raman spectra of these three clusters.

**Figure 3 nanomaterials-12-00414-f003:**
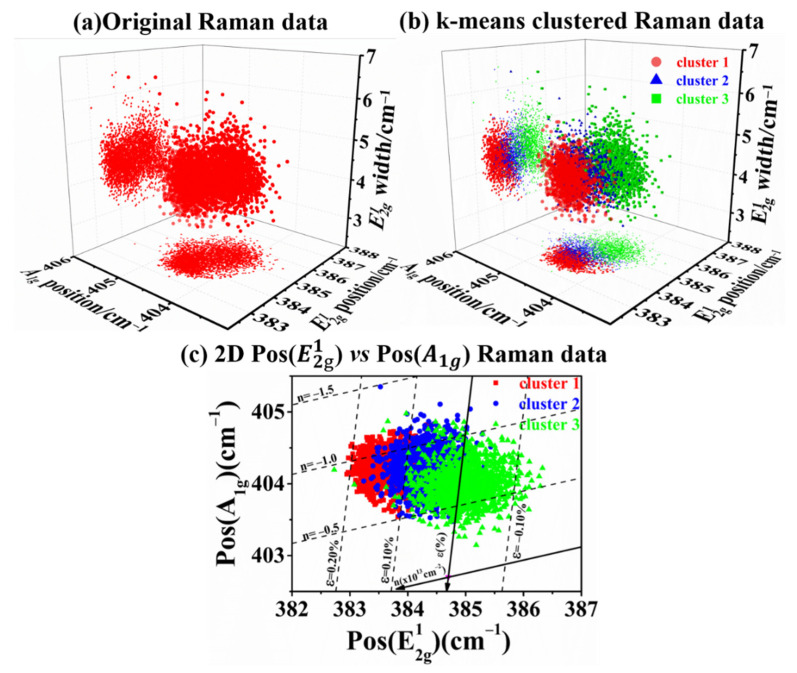
Correlative analysis for Raman spectral features to quantitatively evaluate the inhomogeneity on monolayer MoS_2_. (**a**,**b**) are the 3D correlative graph of peak position (Pos($E_{2g}^{1}$ and Pos($A_{1g}$))) and peak width ($E_{2g}^{1}$) with the original and k-means clustered Raman data, respectively. (**c**) The 2D correlative analysis of Pos($E_{2g}^{1}$) and Pos($A_{1g}$) to accurately extract the contributions of strain and doping effects in cluster 1–3 with red, blue, and green labels.

**Figure 4 nanomaterials-12-00414-f004:**
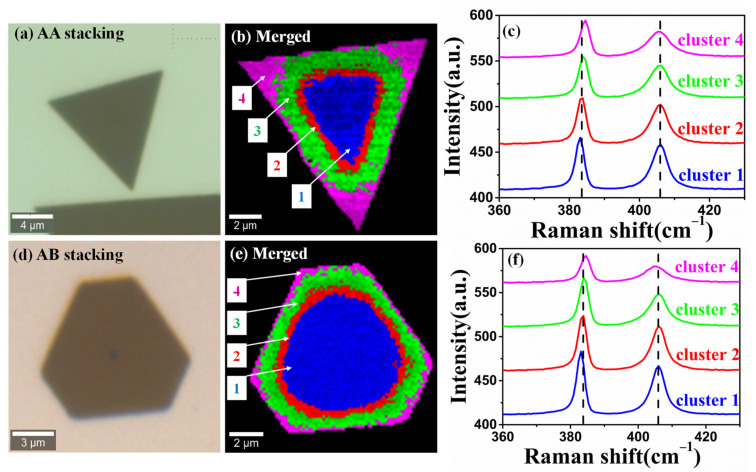
The k-means clustering analysis of bilayer CVD-grown MoS_2_ with different stacking. (**a**,**d**) Optical images of AA- and AB-stacking bilayer MoS_2_. (**b**,**c**) are the color-coded image of four Raman cluster maps and their corresponding mean Raman spectra in AA-stacking MoS_2_, respectively. The related color-coded image and mean Raman spectra for AB-stacking MoS_2_ are shown in (**e**,**f**).

**Figure 5 nanomaterials-12-00414-f005:**
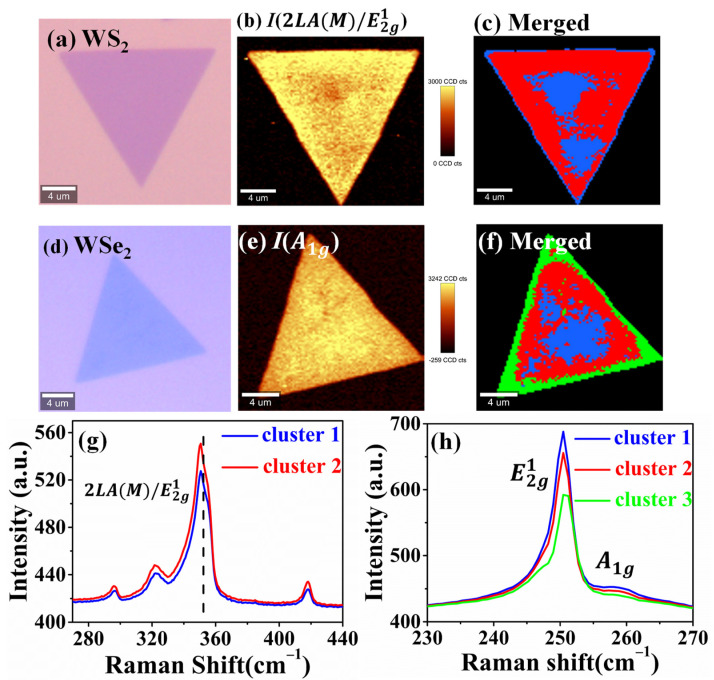
The k-means clustering Raman analysis of monolayer WS_2_ and WSe_2_ on Si/SiO_2_ substrate. (**a**,**d**) the optical images of triangular WS_2_ and WSe_2_ with uniform optical contrast, respectively. (**b**,**e**) The Raman image of the characteristic Raman peak sum intensity ($2LA\left(M\right)/E_{2g}^{1}$ for WS_2_ and $E_{2g}^{1}$ for WSe_2_. (**c**,**g**) Color-coded Raman image with two groups and the correlated mean Raman spectra of WS_2_. (**f**,**h**) Color-coded Raman image and mean Raman spectra of WSe_2_, respectively.

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
