# Peer review of "Direct Detection of Inhomogeneity in CVD-Grown 2D TMD Materials via K-Means Clustering Raman Analysis"

_nanomaterials, 2022, doi:10.3390/nano12030414_

Round 1

Reviewer 1 Report

Xin et al have prepared a highly interesting paper utilizing TMDs for electronic and optoelectronic applications. Several minor changes are needed to make the manuscript worth for publications. My comments are below;

  1. Abstract can be improved in terms of obtained results.
  2. Introduction can be improved. Include some general discussion about other synthesis methods of TMDs materials and how the proposed method is good enough to be applied in the presented work.
  3. Quality of figures can be little improved.
  4. It would be good if authors give subheadings in the results section for clarity.
  5. English language can be polished through out the manuscript.
  6. Repetition of statements should be avoided. 
  7. Include the reference; Trends in Analytical Chemistry, 102 (2018) 75-90.

Author Response

We are grateful for the reviewer’s effort in a careful reading of our manuscript and all the constructive comments. Following the reviewer’s suggestions, we have made proper revisions accordingly. Please find our response in the attachment.

Reviewer 2 Report

The authors address the question of inhomogeneity in CVD-grown 2D
TMD materials. This manuscript provides interesting insights on this matter that have relevance for the use of 2D TMD materials in devices. Therefore author findings will be relevant to a large community. 

Author Response

We are grateful for the reviewer’s effort in a careful reading of the manuscript and we thank the reviewer for all the positive comments on our study.

Reviewer 3 Report

The authors of the paper propose a method to analyze the inhomogeneity of 2D materials through combination of Raman spectroscopy and k-means clustering analysis. The inhomogeneities and their spatial distributions have been obtained in CVD-grown MoS2, WS2 and WSe2 samples. Typical micro-Raman spectral maps of TMDs were measured. The shifts of the characteristic modes induced by mechanical strain and doping was obseved for all samples. However, the proposed k-means clustering analysis is rather sophisticated and does not represent new physical information and sufficient preference in comparison with traditional analysis of Raman intensity and frequency position of vibrational modes in TMDs. To my opinion, the scientific novelty of the paper is limited. 

Author Response

We are grateful for the reviewer’s effort in a careful reading of the manuscript and these constructive comments. We elaborated more on the importance of combining k-means clustering with traditional Raman analysis in introduction section on page 2 and also in the results & discussions part. 

Please find our detailed response in attachement.

Round 2

Reviewer 3 Report

The manuscript was improved after revision. Introduction includes detailed discussion and most appropriated references. More attention was paid to the physical properties of TMDs.

I recommend to publish the manuscript in the present form.